# Anomaly Detection in Multi-Wavelength Photoplethysmography Using Lightweight Machine Learning Algorithms

**DOI:** 10.3390/s23156947

**Published:** 2023-08-04

**Authors:** Vlad-Eusebiu Baciu, Joan Lambert Cause, Ángel Solé Morillo, Juan C. García-Naranjo, Johan Stiens, Bruno da Silva

**Affiliations:** 1Department of Electronics and Informatics (ETRO), Vrije Universiteit Brussel (VUB), 1050 Brussels, Belgium; jlambert@etrovub.be (J.L.C.); angelsm@etrovub.be (Á.S.M.); johan.stiens@vub.be (J.S.); 2Department of Biomedical Engineering, Universidad de Oriente, Santiago de Cuba 90500, Cuba; 3Centre of Medical Biophysics, Universidad de Oriente, Santiago de Cuba 90500, Cuba; jcgnaranjo@uo.edu.cu

**Keywords:** photoplethysmography, PPG, artifact, anomaly detection, multi-wavelength PPG, time series, machine learning, neural networks, supervised learning, unsupervised learning

## Abstract

Over the past few years, there has been increased interest in photoplethysmography (PPG) technology, which has revealed that, in addition to heart rate and oxygen saturation, the pulse shape of the PPG signal contains much more valuable information. Lately, the wearable market has shifted towards a multi-wavelength and multichannel approach to increase signal robustness and facilitate the extraction of other intrinsic information from the signal. This transition presents several challenges related to complexity, accuracy, and reliability of algorithms. To address these challenges, anomaly detection stages can be employed to increase the accuracy and reliability of estimated parameters. Powerful algorithms, such as lightweight machine learning (ML) algorithms, can be used for anomaly detection in multi-wavelength PPG (MW-PPG). The main contributions of this paper are **(a)** proposing a set of features with high information gain for anomaly detection in MW-PPG signals in the classification context, **(b)** assessing the impact of window size and evaluating various lightweight ML models to achieve highly accurate anomaly detection, and **(c)** examining the effectiveness of MW-PPG signals in detecting artifacts.

## 1. Introduction

Health monitoring technologies are evolving, simplifying medical device design and improving accessibility. Wearable devices like smartwatches and fitness trackers are popular for continuously monitoring physiological signals like heart rate and blood oxygen saturation (SpO_2_). They make use of photoplethysmography (PPG), an optical measurement technique. PPG devices consist of a light-emitting diode (LED) to illuminate tissue and a photodetector (PD) to detect reflected photons. By measuring changes in reflected light intensity caused by variations in blood volume, these devices can monitor the pulsatile arterial blood flow.

Over the past few years, increased research interest in PPG technology reveals that, in addition to heart rate, respiration rate, and SpO_2_, other valuable information resides in the PPG pulse shape. Non-invasive blood glucose measurement [1] or blood pressure measurement can be extracted from these signals [2].

Novel PPG devices use a range of wavelengths to capture more detailed and comprehensive information about blood flow in tissue [3]. Longer wavelengths can penetrate deeper into the tissue, enabling the detection of arterial pulsation and changes in blood flow in deeper tissues [4]. On the other hand, shorter wavelengths, such as green light, are preferred for monitoring heart rate during daily activities due to their lower susceptibility to motion artifacts [5]. It has been shown that multi-wavelength photoplethysmography (MW-PPG) improves the accuracy of standard PPG devices for measuring cuffless blood pressure [6] or detecting diabetes [7]. Additionally, a cuffless blood pressure monitor that uses MW-PPG technology was cleared by the US Food and Drug Administration in 2019 [8].

In applications that make use of MW-PPG, it would be beneficial to incorporate anomaly detection techniques to either filter out noisy channels or, alternatively, group channels together to enhance the information available during the anomaly detection process. By doing so, the reliability of the data can be improved for further algorithm processing and clinical decision-making. In addition, this approach can also help identify technical problems with the PPG device, such as sensor malfunction or low signal quality, ensuring prompt maintenance and repair for consistent readings. To achieve these goals, lightweight machine learning (ML) algorithms can be used to identify anomalies in resource-constrained or low-powered environments effectively.

Lightweight ML classifiers refer to machine learning algorithms designed to be computationally efficient and require minimal computational resources. These classifiers are suitable for deployment in resource-constrained embedded systems, such as wearables and IoT devices, where power consumption, memory, and processing capabilities are limited. Examples of lightweight ML classifiers include decision trees, support vector machines, logistic regression, k-nearest neighbors, and neural networks with reduced architectures, such as shallow or sparse neural networks. These classifiers are often optimized for efficiency by using quantization, pruning, and compression of the model parameters. Lightweight ML classifiers are essential for achieving real-time [9], low-power, and accurate anomaly detection in embedded systems [10].

The main objective of this paper is to explore the potential of ML algorithms to improve the reliability of MW-PPG acquisition systems by investigating their use in artifact detection. An overview of several anomalies commonly associated with MW-PPG signals is provided to accomplish this. Different methods for extracting and selecting features from MW-PPG signals are evaluated. Furthermore, a set of relevant features that can be used along with lightweight ML models for artifact detection are proposed.

The main contributions presented in this paper are

A set of features with high information gain in the classification context for anomaly detection in MW-PPG signals are proposed.Evaluation of machine learning models at different window sizes for high-accuracy anomaly detection in MW-PPG.Examination of the effectiveness of combining MW-PPG signals in detecting outliers and filtering anomalous windows from the signal.

The paper is organized as follows: Section 2 reviews related work and state-of-the-art anomaly detection methods in the literature. Section 3 presents a description of PPG signal morphology, the types of artifacts, and the lightweight ML models evaluated in this paper. Section 4 outlines the methodology and the preliminary steps of feature engineering, including the tools, features, and selection methods used. The evaluation of different feature sets and algorithms is presented in Section 5. Finally, Section 6 and Section 7 conclude the paper by completing the proposed methods, models, and their evaluation.

## 2. Related Work

The utilization of ML algorithms for sensor data analysis has witnessed a notable increase in recent times. By applying suitable algorithms to analyze time series data, it is possible to uncover more insights that reside in the data itself and to extract meaningful information from it. The use of ML for sensor data analysis has potential applications in various fields, including healthcare, transportation, and environmental monitoring.

MW-PPG has become an increasingly common technology in various healthcare settings, wearable devices, and smartphones. With the proliferation of PPG sensors, the volume of data generated has increased substantially and the need for accurate and reliable data analysis has become paramount. Moreover, the potential uses of PPG data go beyond healthcare settings. This technology can also be used for monitoring athletes, workers in high-stress environments, and even in everyday consumer devices. With such a vast amount of PPG data being generated, using ML algorithms for anomaly detection and erroneous parameter extraction becomes even more crucial for ensuring the accuracy and reliability of the data.

To assess the quality of PPG signals, optimal methods have been explored in previous studies, such as the one conducted in [11]. In this study, the author compared the effectiveness of seven different signal quality indices (SQI) to the gold standard method, the perfusion index. It has been shown that skewness is a suitable indicator for real-time applications, as it demonstrated the highest performance for window sizes of less than five seconds. In [12], the authors proposed a waveform-based evaluation method that enhanced the rising slope of the filtered PPG signal and correlated it with a pre-determined reference signal during a calibration stage. A nonlinear scaling function transformed the resulting cross-correlation value to an SQI value between 0 and 100. This method achieved a sensitivity of 94.66% and a positive predictive value of 96.78%. In another study [13], a statistical reference model was implemented using the Mahalanobis distance using PPG morphological features to estimate the SQI in continuous heart rate estimation. The set of morphological features included the first derivative, pulse length, and skewness.

In [14], a dynamic time-warping technique was used with a multi-layer perceptron network (MLP) to classify beat-to-beat PPG pulses. This approach achieved high accuracy and sensitivity with a small number of hidden nodes. However, it relies on template matching and does not consider inter-subject variability in the PPG signal. An effective approach for single-channel PPG anomaly detection involves analyzing time-domain and frequency-domain features of truncated signal segments [15]. This method combines DWT and CNN-LSTM for signal reconstruction and applies a threshold-based classification using mean squared error. In [16], the authors proposed a lightweight elliptical envelope algorithm that groups multiple SQIs and classifies the signal as reliable or unreliable. However, the analysis only focused on a one-channel PPG signal and used very large window sizes of 30 s.

All the anomaly detection techniques discussed above are based on template matching or personalized adaptive initialization. The unique characteristics of the signal morphology depend on various factors such as the type of sensor, hardware configuration, or individual traits. This approach is unsuitable for MW-PPG signals due to the impracticality of multiple templates for each channel’s unique morphology.

Therefore, developing lightweight ML algorithms for MW-PPG anomaly detection targeting wearable devices is of utmost importance to enable real-time and accurate detection of artifacts in MW-PPG signals. Such algorithms could be used in wearable devices to increase the measurement’s robustness or mitigate false alarms, which can cause unnecessary anxiety for patients and unnecessary workload for healthcare professionals.

The aim of this paper is to fill the gap in research on anomaly detection in MW-PPG signals, especially for lightweight ML algorithms that can be utilized in portable devices. The study proposes an initial examination of features, comparing two feature extraction methods to identify relevant features, and suggests a set of features that contain data-aware features. Furthermore, it explores the combination of MW-PPG signals for anomaly detection and assesses various lightweight ML models for anomaly detection.

## 3. Background

### 3.1. Photoplethysmogram Signal

The PPG waveform is the resulting light-modulated signal which is captured by the PD and has an inverse relationship with the light absorption in the tissue. The signal can be measured in transmittance or reflectance mode depending on the physical positioning of the light source and the PD. In our case, the reflectance mode is employed, where the sensor and light source are positioned on the same side of the tissue. Figure 1 shows a typical PPG waveform and the most representative features. It can be seen that the PPG pulse has a rising edge that is called the anacrotic phase and a falling edge that is called the catacrotic phase. The first phase indicates the systole, while the second phase indicates the diastole. The dicrotic notch between the two peaks indicates the closure of the aortic valve [17]. The signal consists of a pulsating component (AC) and a non-pulsating component (DC). The former component represents the pulsatile arterial blood and is synchronous and proportional to the subject’s heartbeat [18], whereas the latter is due to tissue parts through which blood does not circulate (bloodless tissue, bone, muscles).

Red, green, and infrared wavelengths can be employed for measurements in the context of MW-PPG. Green light has a shorter wavelength and is more stable in extracting pulse rates [19]. Red light has a longer wavelength and is commonly used to determine arterial oxygen saturation levels alongside infrared light. The latter can penetrate deeper into the tissue, and it is suitable for capturing arterial pulsation [3].

### 3.2. MW-PPG Measurement and Artifact Detection

PPG devices have the potential to incorporate various measurement wavelengths in order to leverage the benefits offered by each channel and create reliable models. Figure 2 depicts different configurations of PPG sensors, ranging from the simplest one-channel system (one PD, one LED) to a more advanced multi-site PPG topology with a total of nine channels (three PDs, three LEDs), where a channel refers to a specific source–detector pair used to measure the PPG signal at a particular wavelength. The inclusion of multiple LEDs and multiple PDs enhances the robustness of the measurement, as artifacts may affect certain channels unevenly while others maintain a high signal quality index [20].

By increasing the number of LEDs and PDs, it becomes possible to minimize the influence of artifacts but also to identify and differentiate artifacts from genuine physiological signals. Moreover, it helps to obtain a comprehensive evaluation of physiological changes by improving feature extraction and analysis. The number of channels can be increased either by including different measurement wavelengths or by using the same wavelength with multiple LEDs. A multi-wavelength approach has the advantage of versatility, noise reduction, different levels of tissue penetration, and enhanced overall information. The MW-PPG configurations shown in Figure 3 can employ two types of anomaly detection:**Channel-level anomaly detection:** each channel is evaluated independently. This method can be adopted when powerful algorithms are used to derive extra information from each PPG channel, such as wave propagation, phase differences, and bilateral differences between different sides of the body. These parameters can hold important information and even be the markers for different diseases such as sclerosis [21] or other arterial diseases [22]. Incorporating an anomaly detection stage before the feature extraction stage can enhance parameter estimation by identifying and discarding anomalous channels. As wearable devices continue to evolve and advance, it is foreseeable that future iterations are expected to integrate more sophisticated methods for detecting complex physiological markers.**Sensor-level anomaly detection:** the channels originating from a single PD are grouped and collectively assessed. The approach can be employed when the overall state of the channels needs to be assessed. By considering multiple channels at the same time, it becomes possible to extract a wider range of features that can be used to more accurately identify anomalies, patterns, or abnormalities in the signal. It can be well-suited for applications that involve long-term PPG monitoring, as well as for better estimating heart rate or SpO_2_ by discarding the windows that contain artifacts.

### 3.3. Types of Artifacts in PPG Signals

An artifact is an error introduced in the measurement process by the measuring equipment or the environment. In a PPG system, artifacts can be introduced into the system by the patient, sensing unit, or the measuring environment.

The quality of the PPG signal depends on the measuring environment, skin properties, temperature, contact force, movements made by the patient during measurement, or ambient light.

To have a better understanding of how the artifacts influence the reflected output light, the Beer–Lambert equation is given, where *I_o_* is the output light intensity, *I_i_* is the input light intensity, ε represents the extinction coefficient, *c_j_* represents the concentration of the tissue, and *d_j_* represents the path length of the reflected light [23]. The tissue can be represented by a model consisting of multiple mediums, denoted by *n*. In this model, the absorbance of light is determined by the combined contributions of all the different absorbers present in the tissue, which may vary in composition and properties [24]:(1)$$I_{o}\left(t\right)=I_{i}\left(t\right)\;\cdot \;exp\left(-\sum_{j=1}^{n}\epsilon_{j}\;\cdot \;c_{j}\left(t\right)\;\cdot \;d_{j}\left(t\right)\right)$$

As shown in Equation (Equation 1), the primary source of artifacts in this context is attributed to the internal deformations of the tissue (*c_j_*) and the spatial displacement of the sensor relative to the skin (*d_j_*). In the presence of motion artifacts, the output intensity can be redefined by considering the primary contributions:(2)$$I_{o}\left(t\right)=I_{i}\left(t\right)\;\cdot \;\gamma_{tissue}\;\cdot \;\gamma_{pulse}\left(t\right)\;\cdot \;\gamma_{position}\left(t\right)\;\cdot \;\gamma_{motion}\left(t\right)$$
where *γ_tissue_* represents the static tissue attenuation, *γ_pulse_* is the component introduced by the pulsatile absorption of the blood, *γ_position_* is due to the change of position, and *γ_motion_* is the component introduced by dynamic changes of the tissue during movement.

Some of the most frequent types of artifacts that have a significant impact on the quality of MW-PPG signals are

**Motion artifacts:** caused by gross body movement, which leads to a change in coupling and distance between the sensor unit and the skin tissue.**Contact force artifacts:** caused by the force magnitude applied over the sensor surface and results in a decrease in the AC/DC ratio of the signal. More details are presented in Appendix A.**Sensor-related artifacts:** malfunction of one sensor component due to physical damage or surface contamination. In an MW-PPG system, due to the increased redundancy of the sensor components, this type of artifact can be detected and discarded.**Environment artifacts:** caused by ambient light or due to a low temperature of the tissue.

Figure 3 shows an MW-PPG signal affected by transient motion artifacts. The highlighted purple area represents the impact of gross finger movement over the surface of the sensor. It can be seen that all the channels are affected simultaneously to varying extents by this type of artifact. The orange area highlights the green channel saturation for a short period due to an abrupt amplitude change. Clipping of one of the channels appears when the measured voltage exceeds the sensitivity range of the PD. It can be noted that the blood pulsation information is still evident in the red and infrared channels.

### 3.4. Machine Learning Techniques

Anomalies in time series data can be detected either using statistical-based methods or more powerful algorithms such as ML algorithms. ML training techniques can be classified into supervised and unsupervised learning. Supervised learning relies on labeled data to train and predict outcomes. Supervised models are more accurate and used for predicting outcomes on new data, while unsupervised models learn inherent features from the training data. Unsupervised learning is employed when extracting insights from extensive datasets or in scenarios where labeled data are not accessible.

In [25], the authors present a vast number of frameworks and ML algorithms that are state-of-the-art and can be easily deployed on wearable devices with the help of open-source frameworks. The majority of models are designed to be compatible with the memory limitations of wearable devices and the timing requirements of real-time applications. The evaluated models encompass various frameworks and configurations, including support vector machines, decision trees, random forests, and shallow MLP networks.

In this analysis, the emphasis is placed on lightweight ML algorithms that are specifically tailored for real-time classification tasks. These algorithms not only offer efficient performance but also possess the capability to be seamlessly ported onto wearable devices, supporting various open-source frameworks.

The paper considers and evaluates the following models:**Decision tree (DT):** a tree-based algorithm that assesses features and traverses branches until a leaf node, which represents a class, is reached.**Random forest (RF):** a collection of decision trees that randomly select specific features and use a voting criterion to determine the output class.**Support vector machine (SVM):** a classical ML algorithm that finds a hyperplane in high-dimensional space to separate examples from different classes. SVM supports linear kernels or different kernel types to create non-linear decision boundaries.**Autoencoder:** an unsupervised neural network that aims to reconstruct the input data. It consists of an encoder–decoder architecture with a bottleneck layer that contains the N-dimensional input vector embedding. It closely resembles an MLP network, where the number of neurons in the output layer matches the number of inputs.

## 4. Materials and Feature Analysis

This section concisely summarizes the techniques utilized for evaluating anomaly detection in MW-PPG signals. Outlining all the steps aims to equip the reader with the necessary knowledge to comprehend and assess the findings presented in Section 5. The current evaluation is outlined in Figure 4, illustrating the overall steps undertaken. Furthermore, the last three stages are presented and discussed in Section 5.

### 4.1. PPG Data Acquisition

The ML models in this work are trained using data recorded from a custom MW-PPG multi-parameter system [26]. This system, as depicted in Figure 5, contains an optical front-end (OFE) and a driver board integrated into a CY8CKIT-059 development module. The OFE includes two PIN-type photodetectors with a sensitivity range of 400 to 1100 nm, along with five LEDs positioned between the PDs. The driver board controls the multiplexing and intensity of the LED lighting. Three specific wavelengths are employed: 525 nm (green), 631 nm (red), and 940 nm (near-infrared). The near-infrared LED is positioned at the geometric center between the two PDs, while two red and two green LEDs are placed in opposition between the PDs. This configuration ensures consistent perception of the three wavelengths by both PDs. The distance between the center of each PD and the near-infrared (NIR) LED is 3.8 mm, while for the red and green LEDs, it is 2.5 mm.

For visible wavelengths, the LED model VLMx1300 series from Vishay Semiconductors is used. For NIR wavelength, the WL-SICW LED from Würth Elektronik is employed. The PD model selected is TEMD5010X01 (Vishay Semiconductors, Malvern, PA, USA) that has a detection area and a relative sensitivity angle of 0.23 mm^2^ and ±60°, respectively. With a spectral sensitivity range of 400 to 1100 nm, the PDs can effectively detect visible and NIR light.

The signals are digitized and transmitted using the CY8CKIT-059 module with an 18-bit resolution and a sampling rate of 125 Hz.

The evaluation proposes to address artifact detection, such as gross body movement or another type of artifact that is not subject-dependent (ambient light, channel saturation). In addition, we included the measured data artifacts that could be considered subject-dependent such as contact force artifacts and low-temperature tissue artifacts [27] that can cause severe signal attenuation.

Table 1 presents details regarding the participants involved in the study, measurement site, and duration. The skin color is determined based on the Fitzpatrick scale, which is a commonly used system to describe skin types in humans.

We use only 3 measurement channels for the current analysis since a sensor unit (1 PD and 3 LEDs) is the base component of various PPG configurations (see Figure 3).

### 4.2. Data Labeling

After the data acquisition process, HeartPy [28], a specialized toolkit developed for analyzing heart rate using PPG data, in conjunction with a custom labeling tool, is utilized to manually label each channel individually. It offers functionalities for various stages of analysis, including signal pre-processing, peak detection, and heart rate computation. One notable feature of HeartPy is the ability to define a normal heart rate interval and mark specific peaks that are deemed incorrect during the analysis process. A 5th order Butterworth low pass filter with a cutoff frequency of 10 Hz is applied to filter out high-frequency noise from the signal [29]. A signal chunk is visually deemed anomalous if it contains significant motion artifacts, irregular waveform shapes, or a low signal quality from which the heart rate cannot be computed. After labeling, the percentage of abnormal signal chunks in the entire dataset is 35%, representing around 1 h and 50 min.

### 4.3. Data Augmentation

Data augmentation is used to increase the size and diversity of the dataset by introducing modifications to the existing data. It is used to simulate channel-wise anomalies such as signal dropout, signal clipping, and other types of sensor-related artifacts. In addition, it is utilized to augment the input data by generating variations of the original signal.

Considering a measurement system with *n* sensors and *m* channels, the output from one channel can be defined as
(3)$$S_{ij}=\left(s_{ij}\left(t_{1}\right),s_{ij}\left(t_{2}\right),\ldots ,s_{ij}\left(t_{k}\right)\right)$$
where i≤n,j≤m. Data augmentation involves applying various transformations or modifications to each measurement channel. The augmenters used are shifting, scaling, warping, baseline drift, noise addition, and dropout.
(4)$$A_{ij}=\left\{\begin{array}{cc} A_{ij}\left(t_{n}\right)=S_{ij}\left(t_{n+w}\right) & \mathrm{where}\;w\;\mathrm{is}\;\mathrm{the}\;\mathrm{shifting}\;\mathrm{factor},\\ S_{ij}⊙s & \mathrm{where}\;s=[s_{1},s_{2},\ldots ,s_{k}]\;\mathrm{is}\;\mathrm{the}\;\mathrm{vector}\;\mathrm{array}\;\mathrm{scaling}\;\mathrm{factor},\\ A_{ij}\left(t_{n}\right)=\phi \left(S_{ij}\left(t_{n}\right)\right) & \mathrm{where}\;\phi :[1,k]\to [1,m]\;\mathrm{is}\;\mathrm{the}\;\mathrm{wrap}\;\mathrm{mapping}\;\mathrm{function},\\ S_{ij}⊕f\left(t\right) & \mathrm{where}\;f\left(t\right),\;t=\left[t_{1},t_{2},\ldots ,t_{k}\right]\;\mathrm{is}\;a\;4\mathrm{th}\;\mathrm{polynomial}\;\mathrm{drift}\;\mathrm{function},\\ S_{ij}⊕N & \mathrm{where}\;N=[n_{1},n_{2},\ldots ,n_{k}]\;\mathrm{is}\;a\;\mathrm{Gaussian}\;\mathrm{noise}\;\mathrm{sequence},\\ S_{ij}⊙M & \mathrm{where}\;M=[m_{1},m_{2},\ldots ,m_{k}]\;\mathrm{is}\;\mathrm{the}\;\mathrm{dropout}\;\mathrm{mask} \end{array}\right.$$

The Python package *Tsaug* [30] is used for generating training samples and new types of channel anomalies. The size of the dataset is increased to approximately 11 h of MW-PPG signals generated using data augmentation. The package offers a customizable pipeline of augmenters for generating clean signals The package also supports batch augmentation for multivariate time series data with channel-wise anomalies. Figure 6 shows an example of MW-PPG time series with synthetic transient dropouts present on one channel. Each element from the dropout mask is generated from a Bernoulli distribution.

Table 2 provides an overview of the MW-PPG dataset, including the duration of each category, the average beats per minute (BPM), and the breathing frequency. For the augmented data, the augmentation is generated on the channel level. This means that at a specific time instant, at least one channel can contain an anomaly.

### 4.4. Feature Extraction and Selection

Before utilizing MW-PPG measurement data and their augmented versions, it is essential to store and pre-process the data to prepare for their application in ML classifiers. Dimensionality reduction techniques like feature extraction and selection are crucial for analyzing high-dimensional data. Feature extraction transforms the input into a lower-dimensional space while preserving relevant information, especially for time series analysis. Feature selection determines the effectiveness of features in relation to the output class.

Before feature extraction and analysis, a windowing process is applied to divide the series into consecutive segments of a fixed length. It has been seen that window sizes between 2 and 5 s are suitable in the context of anomaly detection or activity recognition in PPG time series [31,32]. In the labeling process, the window size used for segmentation is not taken into account. As a result, a window can potentially include samples from both classes. The label of the windows is decided based on a weighted voting function:(5)$$L=\sum_{i=1}^{n}\left(w_{i}\;\cdot \;l_{i}\right)\geq m$$
where $w_{i}$ represents the weight assigned to each window, $l_{i}$ is the binary label of the sample *i*, and *m* is a threshold value. Thus, larger window sizes require a larger number of anomalous samples for the series to be classified as an anomaly.

The *tsfresh* [33] and *TSFEL* [34] software packages are used to extract the features from MW-PPG signals. They offer a wide range of features that capture different aspects of time series data. *TSFEL* provides over 60 features, including spectral, statistical, and temporal characteristics. It has been used in applications such as fault detection in gas turbines [35], monitoring building conditions [36], and activity recognition [37]. *tsfresh* calculates numerous time series features and has been employed in activity recognition [38,39], real-time drilling sensing systems [40], and blood pressure estimation from PPG signals [41].

#### 4.4.1. Feature Extraction

Starting from the dataset that contains only 3 measurement channels (green, red, infrared), a record in the dataset can be seen as a homogeneous multivariate time series:(6)$$S=\left[\begin{array}{cccc} X_{green}\left(t_{1}\right) & X_{green}\left(t_{2}\right) & \ldots & X_{green}\left(t_{k}\right)\\ X_{red}\left(t_{1}\right) & X_{red}\left(t_{2}\right) & \ldots & X_{red}\left(t_{k}\right)\\ X_{ir}\left(t_{1}\right) & X_{ir}\left(t_{2}\right) & \ldots & X_{ir}\left(t_{k}\right) \end{array}\right]$$
where the recorded time series have the same dimension *k* for all the channels.

After applying the windowing process, features are extracted from each window. This operation can be defined as a mapping $\theta_{f}:R^{s}\to R^{l_{f}}$ that captures one specific aspect of a vector of dimension *s* and reduces it to a length $l_{f}$, where $l_{f}\ll s$, *s* is the window size, and $l_{f}$ is the number of attributes in the feature. For a dataset with only three channels, $l_{f}$ is equal to 3. Assuming that $F_{s}$ is a set of features such as $F_{s}=\left\{f_{1},\;f_{2},\ldots ,\;f_{n}\right\}$, the low-dimension result of each mapping function $\theta_{f_{i}}$ is called a *feature*, and it is a characteristic of the time series. The resulting feature vector from one window is $Y_{i}\left(w_{i}\right)\in R^{L}$, where $L=\sum_{i=1}^{\left|F\right|}\left|l_{f_{i}}\right|$, and the final feature matrix of the time series for a number of *n* windows is:(7)$$G=\left[\begin{array}{cccc} Y_{green}\left(w_{1}\right) & Y_{green}\left(w_{2}\right) & \ldots & Y_{green}\left(w_{n}\right)\\ Y_{red}\left(w_{1}\right) & Y_{red}\left(w_{2}\right) & \ldots & Y_{red}\left(w_{n}\right)\\ Y_{ir}\left(w_{1}\right) & Y_{ir}\left(w_{2}\right) & \ldots & Y_{ir}\left(w_{n}\right) \end{array}\right]$$

Giving a binary label vector $L={\left[L_{1},\;L_{2},\ldots ,\;L_{n}\right]}^{T}$, an ML model can be trained on the final feature matrix. In the case of channel-level anomaly detection, a model is trained to have as input only feature vectors of type $Y_{i}\left(w_{i}\right)\in R^{L}$. Thus, only one ML model is tailored to assess all the channels individually, one at a time.

If data fusion is applied at the sensor level, the extracted feature vectors for each channel are grouped together column-wise, and the training feature vector is $Y=[Y_{green}\left(w_{i}\right),\;Y_{red}\left(w_{i}\right),\;Y_{ir}\left(w_{i}\right)]$ in $R^{C}$, where C=3·L. The binary label is set as anomalous if at least one window has this type of label. By considering all three channels together, the model can capture the combined information from different channels. This can potentially provide a more comprehensive representation of the signal and enhance the discriminative power of the extracted features.

#### 4.4.2. Feature Selection

Feature selection is crucial for reducing dimensionality and selecting relevant features in ML algorithms [42]. Including all extracted features can be misleading due to the presence of irrelevant or redundant information. Studies have shown that feature selection can significantly improve classification accuracy in a subset of cases [43]. According to [44], it is often observed that a reduced feature set can achieve higher classification accuracy compared to using the entire feature set. However, the effectiveness of feature reduction depends on the specific data type being analyzed.

During the selection process, feature matrices from all training recordings in the dataset are taken into account. This analysis can be conducted in two ways: either considering all channels together and performing feature aggregation across attributes or conducting separate analyses for each channel individually. Feature aggregation consists in taking the output $\theta_{f}$ of a feature and computing the mean value of all the attributes in the feature for all three channels.

Both *TSFEL* and *tsfresh* have in-built feature selection tests that rely on the class-conditional distribution of each feature. However, comparing class-conditional distributions may not provide a clear numerical ranking or a quantitative measure of feature importance, unlike information gain. The information gain evaluates how each feature contributes to decreasing the overall entropy. The entropy of a discrete random variable *X* with a probability mass function P(X) is defined as
(8)$$H\left(X\right)=E\left[I(X)\right]=E\left[-log(P(X))\right]$$
where *E* is the expected value operator and *I* is the self-information.

An attribute is worthwhile if the information gain with respect to the output class is greater than zero:(9)InfoGain(Attribute,Class)=H(Class)−H(Class|Attribute)

*WEKA* [45], a data mining tool for data engineering projects, is utilized to compute the information gain of each feature in the classification context. The analysis is performed on the training data to ensure that the chosen features generalize well to unseen data and prevent any type of data leakage from the test or validation data. The *InfoGainAttributeEval* method is used to calculate the information gain for each attribute and rank them accordingly.

During feature selection based on information gain, it is common practice to establish a threshold value above zero. Features surpassing this threshold are deemed more relevant and are more likely to be chosen for subsequent analysis or model construction. To automatically determine the optimal number of features for selection, a cut-off threshold selector is employed. *WEKA* supports Recursive Feature Elimination (RFE) by iteratively removing attributes from the feature set to improve the evaluation metric. The search proceeds in a backward manner. When using RFE in combination with an RF classifier, cross-validation is performed to assess the performance of different feature subsets and establish the cutoff threshold. The stop criterion in this context is determined by a heuristic approach, specifically a 2.5% decrease in F1 score compared to the full feature set. This ensures that the retained feature subset maintains an acceptable level of performance. The method is described in Algorithm 1. The function *setInfoGainValues* sets the information gain determined previously with *InfoGainAttributeEval* method. RFE is implemented using *AttributeSelectedClassifier* class from *WEKA*.    
**Algorithm 1:** RF-based RFE feature selection
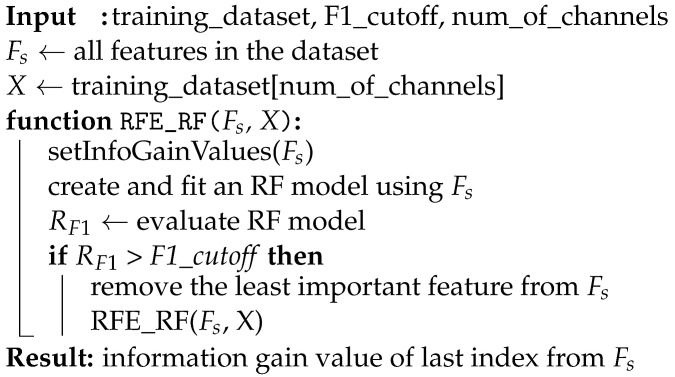


### 4.5. Proposed Features

Feature extraction tools often compute generic features without considering the particularity of input data. To achieve a comprehensive set of features, it is recommended to combine both off-the-shelf and data-aware features. This customization enables the feature set to better align with the unique characteristics and intricacies of the signal, potentially leading to improved anomaly detection performance. Several features are proposed and evaluated in the context of anomaly detection:

**Number of peaks**: represents the maximum amplitude or intensity of the pulsatile signal associated with each heartbeat. It serves as a basic feature or input in combination with other features.

**Peak variance**: offers insights into the stability or consistency of peak occurrences over time. Noise can introduce random fluctuations in the signal, causing the peak values to vary more widely. This increased variability can lead to a higher peak variance as the noise affects the precise location and amplitude of the peaks.

**Peak width variance**: offers insights into the temporal characteristics or dynamics of the peaks in a signal. By analyzing the variability of peak widths, deviations from the expected or normal range can be identified, which may indicate anomalous or abnormal patterns in the signal.

**Mean power spectral density**: one of the methods to determine anomalies in PPG signals is by estimating the signal’s power spectral density (PSD). The power spectrum of a time series describes the distribution of power into frequency components composing that signal. It has been seen that a mean magnitude of the power density between 50 and 200 Hz has good results in separating normal windows from abnormal ones.

**Frequency bands slope**: by calculating the slope of the PSD curves between the low-frequency and high-frequency ranges, one can quantify the transition or change in power between these frequency bands. More details about this feature are shown in Appendix C. The normal frequency range for heart rate activity is typically 1–2 Hz. Studies have shown peaks at 0.3 Hz (respiratory frequency) and 1.7 Hz (heart rate) in the FFT analysis of PPG signals [46]. Another study identified peaks at 0.19 Hz (respiratory frequency) and 1.44 Hz (cardiac pulse) [29]. Comparing the signal components within the expected heart rate range (1–2 Hz) to higher frequencies (>3 Hz) reveals higher slope values for normal windows and lower ratios for anomalous windows.

## 5. Results

The MW-PPG time series are segmented into fixed-size windows to ensure that the ML model receives comprehensive data. The windowing procedure has a 50% overlap between adjacent windows [47]. This also brings the advantage of a higher number of training samples. The dataset is divided such that 10% of the windows are set aside for testing, another 10% for validation, and the remaining 80% are allocated for training. The number of training, test, and validation windows is determined by the window size used during the segmentation of the signals, leading to variations in their respective counts as shown in Table 3.

### 5.1. Determining the Feature Sets

Creating a well-designed and informative feature set is crucial to enhance the model’s ability to distinguish between different classes or make precise predictions. The window size used for the analysis has a length of 5 s. The feature extraction step is performed using both *TSFEL* and *tsfresh*, followed by a pre-processing step to handle missing values or any other data quality issues. Finally, normalization of the features is necessary to ensure that they are on a comparable scale.

In the feature selection analysis, two methods are compared and evaluated. The combined-channels approach involves feature aggregation of all channels when assessing the information gained. It captures collective information from all channels to assess feature relevance. The second method involves conducting separate analyses for each channel individually. The purpose of this comparison is to determine the most effective final feature sets.

The individual channel approach reveals that treating each channel independently results in a set of features that have a decreased information gain. In Figure 7, it can be observed that when all the channels are grouped together during the relevance test, the extracted set of relevant features has an increased overall information gain with respect to the output class. Thus, selecting the most relevant subset of features reduces the noise from the data and improves the classification metrics.

Through the RF-based RFE algorithm, an information gain threshold of 0.2 is determined for the *tsfresh* feature set, while a threshold of 0.15 is determined for the *TSFEL* feature set.

The proposed features described above are evaluated in terms of information gain. The values are presented in Table 4 and surpass the threshold values established for the two other feature sets. A new custom feature set is created by combining features extracted from *tsfresh* and *TSFEL*, along with the proposed features. The selection of features with the highest information gain from all the feature sets is performed using the *BestFirst* greedy algorithm from WEKA.

The most relevant features identified for both *TSFEL* and *tsfresh*, along with the custom feature set, are shown in Table 5. The *TSFEL* feature set consists of 15 features, while the *tsfresh* feature set comprises 13 features. For the custom feature set, the number of relevant features is reduced to 10. A detailed analysis of the underlying structure and variability within each window type for each feature set can be found in Appendix B.

When sensor-level anomaly detection is used, a specific feature should be computed from all three channels. Thus, the relevance of each channel-specific feature is determined such that redundant information among the channels is removed. When channel-level anomaly detection is employed, the features are computed from each channel and evaluated one at a time. In this case, the number of feature attributes is equal to the number of features. In the sensor-level approach, the number of attributes for *TSFEL* and *tsfresh* feature sets is 21, while for the custom feature set, it decreases to 16. This suggests that the custom feature set may capture the most relevant and distinctive information, resulting in a more concise representation with a decreased number of attributes. Table 5 provides a summary of the final feature sets along with channel-wise feature attributes that are used during sensor-level anomaly detection.

### 5.2. Window Size Analysis and ML Evaluation

This section focuses on investigating the impact of different window sizes on input data and evaluating the performance of channel-level and sensor-level anomaly detection. The models are analyzed using various window sizes, namely 2, 4, 6, 8, and 10 s. Determining the optimal window size leads to improved anomaly detection algorithms, increased accuracy in identifying anomalies, and decreased false positive rates.

Performance evaluation of the models involves the use of metrics such as accuracy and F1 score. Accuracy provides a general measure of the model’s ability to correctly classify normal and abnormal data points. On the other hand, the F1 score takes into account both precision and recall, offering a more balanced evaluation of the model’s performance.

Grid search cross-validation is employed to optimize the hyperparameters of the models. The final hyperparameters obtained through this process are as follows:**DT**: Gini impurity is used as a splitting criterion; the maximum depth is set to 15.**RF**: Gini impurity is used as a splitting criterion; a maximum depth of 10 is set, and there are a total number of 10 estimators.**SVM**: uses a Gaussian kernel with a low gamma value of 0.0001; probability estimates are enabled; the misclassification penalty is set to 1.**Autoencoder**: employs four fully connected layers, where the hidden layers have five neurons and use the ReLu activation function; the encoder/decoder has an input size equal to the length of the feature vector based on the feature set used or on the anomaly detection type.

#### 5.2.1. Channel-Level Anomaly Detection

The performance metrics for all the window sizes and classifiers are shown in Table 6 and Table 7. Moreover, all the feature sets are evaluated on each classifier. For the autoencoder, besides the feature sets described earlier, the model is also evaluated on the PPG time series itself, allowing the autoencoder to learn representations directly from the raw data.

As shown in Table 6, the impact of window size on performance metrics shows a consistent trend. Very short or very long windows decrease the performance, while the optimal window size ranges from 6 to 8 s. The interval captures relevant information without compromising the accuracy of MW-PPG anomaly detection. Adopting medium lengths between 6 s and 8 s is recommended for improved anomaly detection across classifiers and feature sets.

The autoencoder, when used with the custom feature set, achieves a higher accuracy and F1 score compared to other classifiers. For a window size of 4 s, the autoencoder outperforms the other classifiers. A difference of 2 s can be crucial for systems that are time-constrained and cannot sample the signal for such high intervals; or for systems in which high responsiveness is needed. Comparing all feature sets, using feature extraction provides more discriminative information and a more concise representation of the PPG data compared to using raw data.

Upon comparing the performance of different feature sets, it becomes evident that the custom feature set consistently achieves competitive results across most classifiers and window sizes. Moreover, one advantage of the custom feature set is that the accuracy and F1 score increase for window sizes outside the optimal interval of 6 or 8 s. Firstly, smaller window sizes (2 and 4 s) facilitate real-time monitoring and analysis of PPG signals. This advantage becomes particularly significant when there is an increase in the number of channels as it enables immediate feedback and facilitates efficient anomaly detection. Secondly, larger window sizes (8 and 10 s) might be suitable for applications where computational resources or energy efficiency is a must. One example could be a measurement device that sends only artifact-free data to the cloud. Reducing the frequency of processing leads to more efficient resource utilization.

#### 5.2.2. Sensor-Level Anomaly Detection

Adopting sensor-level anomaly detection depends on the specific characteristics and requirements of the application. The analysis is performed only over the custom feature set since it obtained the highest performance, and it has been shown how data-aware features improve the performance metrics for all window sizes for channel-level anomaly detection.

In Figure 8, the gain in terms of accuracy and F1 score compared to a channel-level anomaly detection are shown. There is a substantial performance increase for small window sizes under 5 s when the channels from one sensor unit are grouped together during classification. However, it is worth noting that this approach presents some challenges for higher window sizes. For larger window sizes, it may be necessary to use more sophisticated models to capture the complex relationships between different channels. Only the autoencoder keeps a positive gain in performance metrics for window sizes higher than 5 s.

The results suggest that in MW-PPG applications where high responsiveness, more fine-grained anomaly detection, or higher temporal resolution is needed, it is recommended to adopt a sensor-level anomaly approach and to combine the channels for better artifact detection.

## 6. Discussions

Our evaluation demonstrates the advantages of utilizing multi-wavelength measurements in PPG devices for detecting anomalies, whether at the channel-level or sensor-level. This is achieved by leveraging a feature set consisting of relevant features. Both methods can be employed depending on the application characteristic and requirements. Nonetheless, devices that already make use of MW-PPG technology to measure SpO_2_ and heart rate can take advantage of the multichannel measurement to avoid erroneous parameter estimation by utilizing a sensor-level detection stage in its processing chain. Moreover, the results suggest that the transition from one-wavelength or one-channel devices, as shown in Figure 3, apart from enhanced signal quality and more comprehensive physiological information, brings the advantage of increased sensitivity in artifact detection.

The evaluated models are designed to be lightweight and suitable for wearable devices. Therefore, the decrease in accuracy for higher window sizes is not considered a drawback. Smaller window sizes are preferred for real-time applications and resource-constrained devices. This choice reduces memory consumption for storing input signals and improves system responsiveness. Further analysis of classifiers’ memory footprint and inference time for both approaches and different window sizes should be performed. Comparing our implementations with other lightweight anomaly detection methods, the models and features proposed in the current paper outperform the ones presented in [16] regarding F1 metrics and the window size. Even if the mentioned implementation achieves an F1 score of 0.97 for channel-level classification, it does this at the expense of window sizes that are six times larger than the assessed window sizes in our approach.

Comparing our implementations with other anomaly detection approaches, namely deep-learning-based approaches, that are not intended to be lightweight and target GPU processors [15], our implementation obtains comparable results for window sizes between 3 and 6 s in terms of detection rate at the channel level.

The combined channel strategy’s scalability is exciting for future research, especially for increasing numbers of MW-PPG channels. However, a challenge with the increased number of channels is the complex feature space, making training lightweight models harder. More research is necessary to determine the scalability of MW-PPG systems in real-world settings.

However, it is important to acknowledge that the small size of our subject group may limit the generalizability of the results. Anomaly detection algorithms heavily rely on learning patterns and detecting deviations from those patterns. The evaluated algorithms may not have sufficient diverse data to accurately capture the full range of motion artifacts and contact force variations that can occur across different individuals. To mitigate this limitation, we applied data augmentation techniques, such as scaling, warping, and noise addition, to introduce additional instances of artifacts based on the available measured data. This approach aimed to expand the dataset and provide a more diverse representation of potential anomalies.

One limitation of this study is that only one sensor configuration is exploited. Different sensor configurations can exhibit variations in terms of placement and sensitivity and can result in different types and magnitudes of artifacts. The evaluated anomaly detection algorithms and feature sets may be optimized to effectively handle the specific artifacts and challenges associated with the current configuration but may not perform as well when applied to different sensor configurations.

## 7. Conclusions

MW-PPG has revolutionized PPG devices and enabled non-invasive assessment of cardiovascular diseases with resource-constrained devices. Anomaly detection is necessary for accurate and reliable MW-PPG devices but poses significant challenges. Our analysis evaluates the combination of MW-PPG signals and the performance of lightweight ML algorithms for accurate anomaly detection using different feature sets and window sizes. The evaluation of MW-PPG signal features, along with the incorporation of data-aware proposed features, leads to improved accuracy across various window sizes. Additionally, the selected feature set and window sizes allow for a comparative analysis of multiple ML algorithms for anomaly detection, both at the channel-level and sensor-level.

The results show that a multichannel approach outperforms individual channels for anomaly detection, and our results show that the highest accuracy is obtained with shorter window sizes. Compared with state-of-the-art anomaly detection techniques in PPG signals, our results outperform the equivalent ML anomaly detectors and obtain comparable results to deep learning techniques.

## Figures and Tables

**Figure 1 sensors-23-06947-f001:**
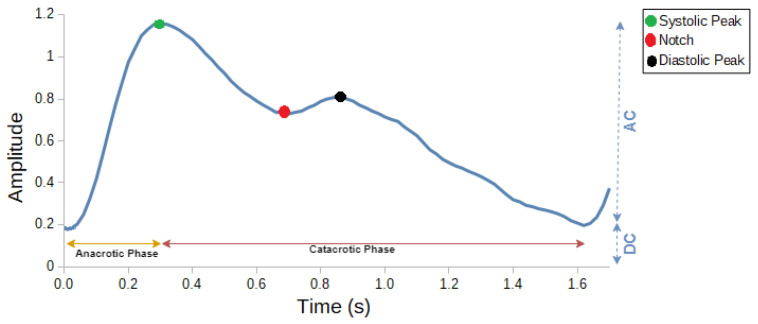
Typical PPG waveform morphology, with systolic peak, diastolic peak, and dicrotic notch highlighted.

**Figure 2 sensors-23-06947-f002:**
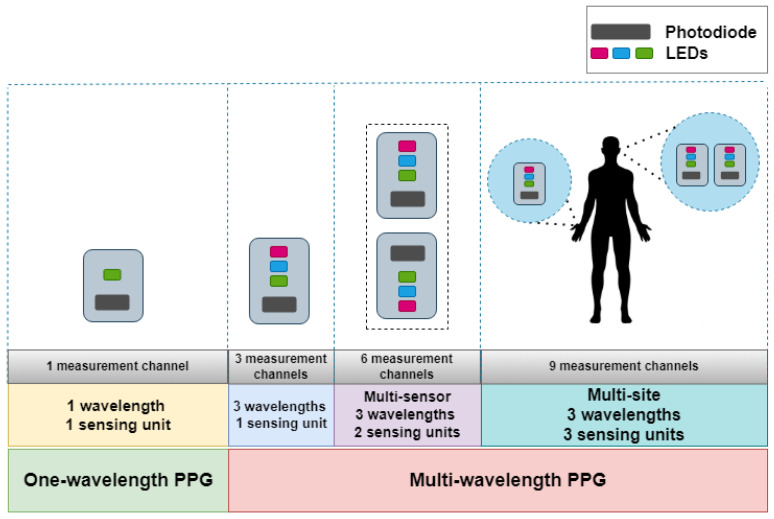
Various configurations of PPG measurement devices.

**Figure 3 sensors-23-06947-f003:**
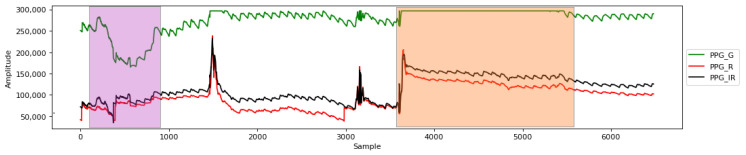
Types of artifacts in an MW-PPG measurement. Purple area: body movement. Orange area: green channel saturation.

**Figure 4 sensors-23-06947-f004:**
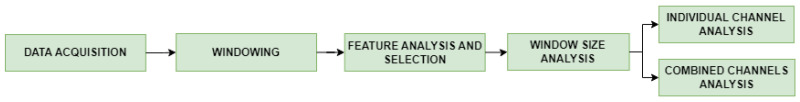
Methodology steps followed for evaluating ML anomaly detection in MW-PPG signals.

**Figure 5 sensors-23-06947-f005:**
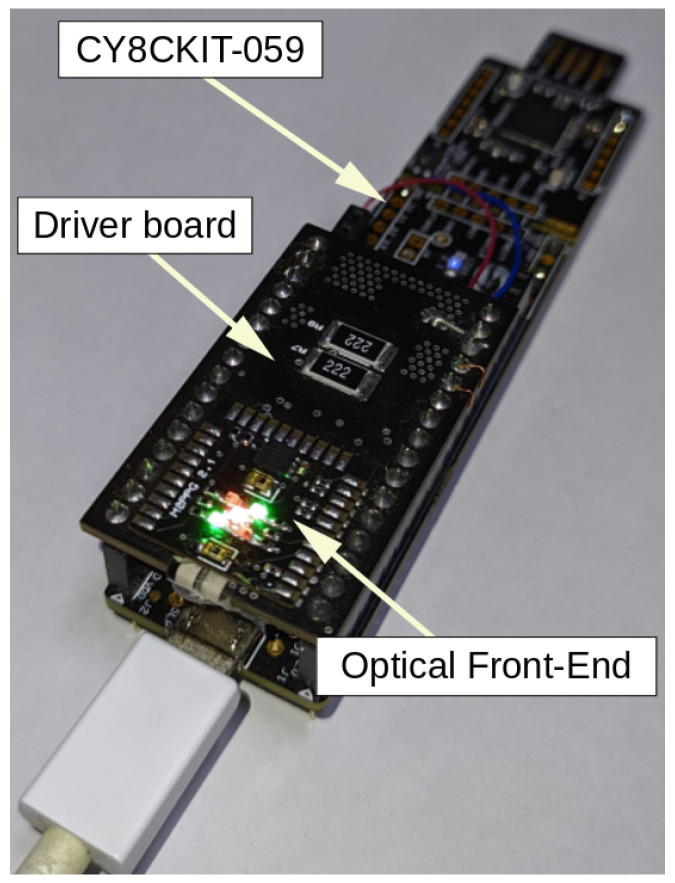
Multi-wavelength front-end and driver board used for data acquisition.

**Figure 6 sensors-23-06947-f006:**
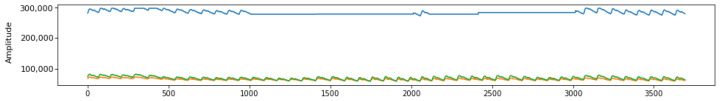
Sensor-related artifact: PPG signal dropout. Synthetically generated artifact.

**Figure 7 sensors-23-06947-f007:**
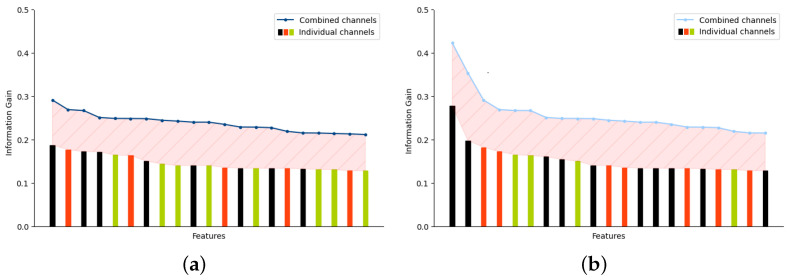
Information gain difference between relevant features determined taking into account all the channels vs. relevant features determined for each individual channel. (**a**) Features extracted using *tsfresh*. (**b**) Features extracted using *TSFEL*.

**Figure 8 sensors-23-06947-f008:**
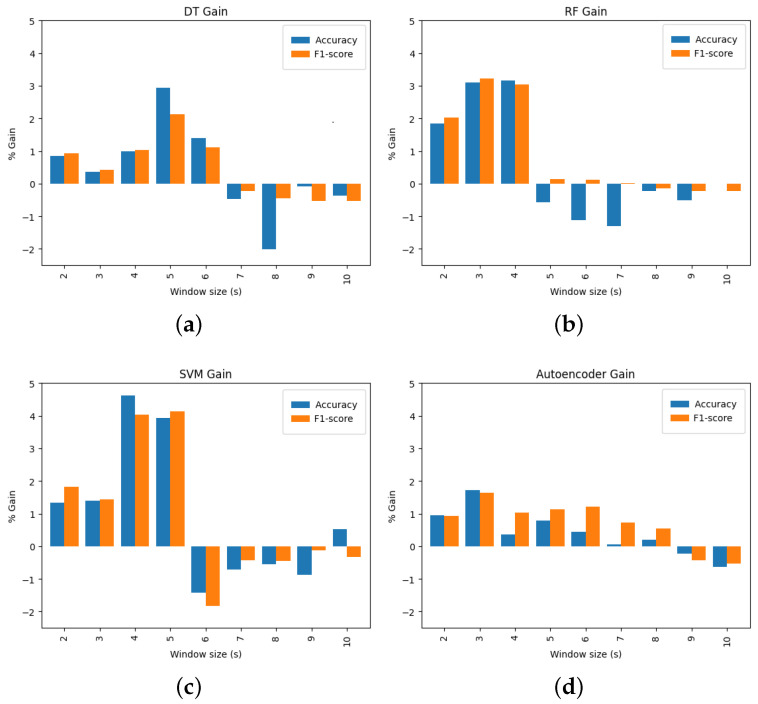
Accuracy and F1 score gains across classifiers and window sizes when sensor-level anomaly detection is adopted. (**a**) Decision tree (**b**) Random forest (**c**) SVM (**d**) Autoencoder.

**Table 1 sensors-23-06947-t001:** Dataset details regarding participants involved.

	Gender	Age	Skin Color	Data Duration (Minutes)	Measurement Site
**Subject I**	Male	24	Type III	115	Left index finger
**Subject II**	Male	26	Type II	145	Left index finger

**Table 2 sensors-23-06947-t002:** Summary of the final dataset used for anomaly detection in MW-PPG.

Data Information			Data Duration (Minutes)				
**Breathing** **Frequency (Hz)**	**Average BPM**	**Measurement Channels**	**Clean**	**Motion Artifacts**	**Contact Force Artifacts**	**Sensor Artifacts**	**Environment Artifacts**
0.172 ± 0.08	91.6 ± 8.9	Green, Red, Infrared	150	50	30	10 (on Green channel)	20
		**Augmented**	60	50 (channel level)	70 (channel level)	200 (channel level)	x

**Table 3 sensors-23-06947-t003:** Number of training, test and validation samples for each window size.

Window (s)	2	4	6	8	10
**Training**	47,880	21,280	15,960	10,640	8512
**Test/Validation**	6320	3660	2775	2330	2064

**Table 4 sensors-23-06947-t004:** Information gain of the proposed features.

Feature	Peak Variance	Frequency Bands Slope	PSD Mean	Number of Peaks	Peak Width Variance
**Information Gain**	0.42233	0.35332	0.21323	0.20582	0.20355

**Table 5 sensors-23-06947-t005:** Final feature sets. Channel-wise feature attributes. Ch1—green, Ch2—red, Ch3—infrared.

TSFEL Feature Set		*tsfresh* Feature Set		Custom Feature Set	
**Feature Name**	**Channels**	**Feature Name**	**Channels**	**Feature Name**	**Channels**
neg_turning_points	ch3	binned_entropy	ch2, ch3	peak_var	ch2, ch3
pos_turning_points	ch3	change_quantiles_14	ch3	frequency_bands_slope	ch1, ch2
wavelet_entropy	ch1, ch2, ch3	change_quantiles_16	ch3	neg_turning_points	ch3
median_diff	ch2, ch3	change_quantiles_18	ch3	pos_turning_points	ch3
median_absolute_diff	ch3	change_quantiles_17	ch1	binned_entropy	ch2, ch3
entropy	ch1, ch2, ch3	cid_ce_normalize	ch1, ch2, ch3	wavelet_entropy	ch1, ch2, ch3
MFCC_3	ch1, ch2	lempel_ziv_complexity	ch2, ch3	PSD_mean	ch2
MFCC_2	ch2	change_quantiles_15	ch3	num_of_peaks	ch1
MFCC_7	ch1	agg_linear_trend_5	ch1, ch2, ch3	peak_width_var	ch2
MFCC_6	ch2	change_quantiles_19	ch2	median_diff	ch2, ch3
MFCC_1	ch2	change_quantiles_20	ch3		
median_frequency	ch3	reoccurring_datapoints	ch1, ch2, ch3		
FFT_mean_coefficient_130	ch2	change_quantiles_10	ch1		
spectral_kurtosis	ch3				
spectral_skewness	ch2				
	**Total: 21**		**Total: 21**		**Total: 16**

**Table 6 sensors-23-06947-t006:** Evaluation of performance metrics for various input feature sets, window sizes, and classifiers.

Window	Decision Tree						Random Forest						SVM					
	TSFEL		*tsfresh*		Custom		TSFEL		*tsfresh*		Custom		TSFEL		*tsfresh*		Custom	
	**Acc**	**F1**	**Acc**	**F1**	**Acc**	**F1**	**Acc**	**F1**	**Acc**	**F1**	**Acc**	**F1**	**Acc**	**F1**	**Acc**	**F1**	**Acc**	**F1**
**2 s**	0.85	0.73	0.85	0.75	0.93	0.85	0.88	0.72	0.88	0.73	0.93	0.83	0.83	0.75	0.87	0.71	0.92	0.82
**4 s**	0.91	0.86	0.92	0.83	0.92	0.89	0.95	0.81	0.93	0.84	0.96	0.85	0.98	0.88	0.92	0.82	0.90	0.81
**6 s**	0.90	0.88	0.87	0.82	**0.96**	**0.94**	0.91	0.82	0.87	0.82	**0.98**	**0.88**	0.90	0.79	0.84	0.80	0.91	0.83
**8 s**	0.82	0.84	0.83	0.81	0.93	0.90	0.87	0.75	0.84	0.80	0.95	0.88	0.87	0.74	0.82	0.77	**0.95**	**0.86**
**10 s**	0.79	0.76	0.80	0.76	0.97	0.90	0.81	0.73	0.81	0.80	0.98	0.85	0.83	0.71	0.81	0.72	0.93	0.85

**Table 7 sensors-23-06947-t007:** Autoencoder results of accuracy and F1 score for different input feature sets and window sizes.

Window	Raw Signal		TSFEL		*tsfresh*		Custom	
	**Acc**	**F1**	**Acc**	**F1**	**Acc**	**F1**	**Acc**	**F1**
**2 s**	0.87	0.64	0.91	0.74	0.90	0.73	0.92	0.77
**4 s**	0.90	0.67	0.95	0.89	0.94	0.86	**0.99**	**0.98**
**6 s**	0.92	0.68	0.94	0.87	0.92	0.82	0.97	0.96
**8 s**	0.88	0.63	0.94	0.85	0.92	0.82	0.96	0.93
**10 s**	0.81	0.61	0.90	0.82	0.88	0.81	0.91	0.88

## Data Availability

Data will be available under request.

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
