# Peer review of "Anomaly Detection in Multi-Wavelength Photoplethysmography Using Lightweight Machine Learning Algorithms"

_sensors, 2023, doi:10.3390/s23156947_

Round 1
Reviewer 1 Report
The manuscript reports experiments and analysis of lightweight anomaly detection algorithms photoplethysmography.
The manuscript has a lot of descriptive content and it is also very long to comprehend the research work done on anomaly detection in multi-wavelength PPG technology. 26 pages are many for a paper that simply covers a few ensemble learning methods, such as decision trees, Random Forest, SVM, a custom SVM algorithm, and an autoencoder towards machine learning algorithms, and that are state-of-the-art for the use case, in a top quality journal such as Sensors! The F1 scores and accuracy for the custom SVM algorithm are impressively high for various feature sets experimentally investigated in the study and these results can be interesting in the context of use case that use hand crafted feature generation based classifiers.
The authors needs to shorten the manuscript by focusing the main experiments with the selected anomaly detection algorithms and present the most striking results in the main paper to 16 pages. Anything that does not fit in 16 pages can be made available online as Supplemental Material online.
In addition the English and presentation style of the manuscript should be improved, for example using some concept diagrams, more equations, and succinctly described methods.
Reviewer 2 Report
This paper presents a study of anomaly detection algorithms to detect various types of artifact in multiple wavelength PPG signals. Using a limited true dataset with data augmentation, the authors describe the identification of salient features and the training and testing of various ML models to separate artifactual time windows from clean time windows of data.
I believe this paper could benefit from several major and a few minor revisions, as detailed below.
Major:
1. Throughout the paper, the terms "multi-wavelength", "multi-channel", "multi-site", and "multi-sensor" are used casually without clear delineation or definition. This is very confusing to understand and should be distinguished early on clearly. For example, are there multiple sensors for each wavelength as well as multiple channels per sensor?
2. How was the data collected? Was there IRB approval? What were the age ranges of the subjects etc.? Was it done fully in a controlled setting? How was true artifact annotated? None of this is included.
3. How was ground truth of artifact determined in the data? Was it done by visual inspection? Was it done for each channel? What were the criteria? In what window size? This was left unclear throughout.
4. The data augmentation process needs to be explained in more detail.
5. Why were the tsfresh and TSFEL packages chosen to be used in the first place?
6. Is all of Section 4 done using the entire dataset? Or just the training data? This is not clear. If Section 4 is performed on the entire dataset, does this not introduce bias into Section 5's analysis since the features were selected by including the test data?
7. Are the y-axis labels in Figures 12-14 Entropy or Information Gain?
8. The threshold setting process used to arrive at a subset of features needs to be better explained.
9. The motivation for the entire PCA section (Section 5.2) is not clear or compelling, and this analysis may be better served in the Supplementary Material. Also it is not clear if the PCA was performed separately on normal and anomalous windows or altogether.
10. The terms "custom" and "proposed" features are both used and the delineation is unclear.
11. Can the authors provide more clarity on what the features in each of the final datasets in Section 5 actually consist of? How many features are in each feature set and how are they defined? Otherwise, it is difficult to reproduce these results.
12. Are all of the results reported in Section 5 on the test set? Can the number of windows in the training, validation, and test sets be noted for each window size in a table?
13. What about F1 score in Figure 22?
14. There is no discussion of limitations of the study in Section 6. For example, one major one is that this only includes data from 2 subjects.
Minor:
1. Define 'lightweight' ML earlier in the paper. Currently the definition is in Section 3.3 but the term is referenced in Section 2 as well.
2. Can the authors also label the AC and DC components of the signal in Figure 1?
3. The reference to [16] towards the end of Section 2 seems placed a bit late in the section. It should be moved up a few paragraphs earlier when all of the other previous studies are mentioned.
4. Figure 16 needs an x-axis label.
5. Consider renaming LF/HF ratio since that has a very specific definition in the very related domain of heart rate variability.
Reviewer 3 Report
The study focus is to test effect of temporal window sizes, signal feature sets and different ML algorithms (DT, RF, SVM, and RCA) on accuracy and F1 of the anomaly detection in MW-PPG signals. The study is interesting and has applications especially in wearables where light weight and optimized algorithms are needed for PPG analysis.
The authors identified the optimal signal feature sets, window sizes for different algorithms and demonstrated that simultaneous analysis of multichannel signal outperforms single channel signal analysis.
The article is well written and includes all needed information to understand the study and its results. However, the article is a bit lengthy therefore I’d suggest to reduce it by moving the non-necessary content to Appendix or removing it completely. There are many repetitions, some figures are not needed, and some well known algorithms or approaches are unnecessarily explained in details. An interested reader without proper background would easily find needed information in the literature.
Other comments are:
Introduction
- P.1 par2: “other valuable information resides in the PPG pulse shape” Please write which additional info can be extracted from PPG signals (e.g. respiration rate)
Section 3
- The descriptions of PPG artifacts, ML algorithms and evaluation metrices are very detailed, which is an advantage for a newcomer in the field. While they are unnecessary for experienced reader. I’d suggest the authors to reduce the section to the minimum necessary information, while the extensive descriptions can be moved to Appendix.
- P.8 par. 4: RCA – write full name, “robust collaborative autoencoder”
Section 4
- P9. Par. 3: Provide information about models and manufacturers, actual distances between LEDs and LDs, signal acquisition.
- P9. Par. 4: “recorded data measured from two subjects” – Provide more information about the subjects (skin color, age, gender). Including only two subjects does not cover population variation, therefore the study results should be discussed considering the small sample size. A statement about the medical ethics committee approval is missing, since the study was performed on human subjects.
- P.9 par. 4: Please provide information how the artifacts were introduced to the signal.
- P. 15: “Peak variance” calculation – How noise effects the peak location determination?
- Also, this section is very detailed. Please reduce the length of it and keep only important figures (e.g., Fig.10 and 11 can be removed)
- P. 15: “Low-Frequency/High=Frequency (LF/HF) ratio” – “=” symbol should be replaced by “-”
- P.15 par. 4: It is still not clear how LF/HF is calculated. Is this the ratio between the power density amplitudes at low and high frequencies? What does “slope” mean?
- P.16 Fig 16: What is the x-axis? Is it a window number? What are the features in the graphs (peaks/dips)?
- P.16: “Number of peaks” is not explained.
- TSFEL and tsfresh feature sets are not explained
Section 5
- P.17 par. 4: The “random forest” method is described two times, but once it should be the “decision tree” method.
- P.21 Table 2: Explain the difference between custom feature set and the other two.
Discussion
- The authors should discuss effect of a very small subject group size (two volunteers) on the study results. How does it affect them? What would they expect if a darks or obese volunteer would participate?
- How the results of this study would impact the wearables design?
